# Mitochondrial Oxidative Stress Is the General Reason for Apoptosis Induced by Different-Valence Heavy Metals in Cells and Mitochondria

**DOI:** 10.3390/ijms241914459

**Published:** 2023-09-22

**Authors:** Sergey M. Korotkov

**Affiliations:** Sechenov Institute of Evolutionary Physiology and Biochemistry, Russian Academy of Sciences, Thorez pr. 44, 194223 St. Petersburg, Russia; sergey-korotkov@mail.ru or korotkov@iephb.ru

**Keywords:** heavy metals, apoptosis, oxidative stress, toxic mechanisms, mitochondrial permeability transition pore

## Abstract

This review analyzes the causes and consequences of apoptosis resulting from oxidative stress that occurs in mitochondria and cells exposed to the toxic effects of different-valence heavy metals (Ag^+^, Tl^+^, Hg^2+^, Cd^2+^, Pb^2+^, Al^3+^, Ga^3+^, In^3+^, As^3+^, Sb^3+^, Cr^6+^, and U^6+^). The problems of the relationship between the integration of these toxic metals into molecular mechanisms with the subsequent development of pathophysiological processes and the appearance of diseases caused by the accumulation of these metals in the body are also addressed in this review. Such apoptosis is characterized by a reduction in cell viability, the activation of caspase-3 and caspase-9, the expression of pro-apoptotic genes (*Bax* and *Bcl-2*), and the activation of protein kinases (ERK, JNK, p53, and p38) by mitogens. Moreover, the oxidative stress manifests as the mitochondrial permeability transition pore (MPTP) opening, mitochondrial swelling, an increase in the production of reactive oxygen species (ROS) and H_2_O_2_, lipid peroxidation, cytochrome c release, a decline in the inner mitochondrial membrane potential (ΔΨ_mito_), a decrease in ATP synthesis, and reduced glutathione and oxygen consumption as well as cytoplasm and matrix calcium overload due to Ca^2+^ release from the endoplasmic reticulum (ER). The apoptosis and respiratory dysfunction induced by these metals are discussed regarding their interaction with cellular and mitochondrial thiol groups and Fe^2+^ metabolism disturbance. Similarities and differences in the toxic effects of Tl^+^ from those of other heavy metals under review are discussed. Similarities may be due to the increase in the cytoplasmic calcium concentration induced by Tl^+^ and these metals. One difference discussed is the failure to decrease Tl^+^ toxicity through metallothionein-dependent mechanisms. Another difference could be the decrease in reduced glutathione in the matrix due to the reversible oxidation of Tl^+^ to Tl^3+^ near the centers of ROS generation in the respiratory chain. The latter may explain why thallium toxicity to humans turned out to be higher than the toxicity of mercury, lead, cadmium, copper, and zinc.

## 1. Introduction

*Early research on heavy metal toxicity*. This section is dedicated to a summary of heavy metal toxicity studies that have been performed in previous years. This summary will allow the reader to trace the evolution of the efforts of various researchers on this issue and to understand why the emphasis has shifted from the research of isolated mitochondria to the study of various cellular systems in vitro and in vivo.

**Ag(I).** It was previously shown that intense rat liver mitochondria (RLM) swelling was induced after the reaction of Ag^+^ with 12% rapidly reacting SH groups of the inner membrane [1]. Ag^+^ inhibited Na^+^, K^+^-ATPase [2], and mitochondrial malate dehydrogenase from bovine brain [3]. Ag^+^ increased K^+^ efflux, coincident with net Ca^2+^ uptake, and it stimulated the ouabain-sensitive oxygen consumption rate in experiments with rabbit renal cortical tubule suspensions [4,5]. Thiol reagents (n-ethylmaleimide (NEM), dithiothreitol (DTT), and reduced glutathione) reversed these Ag^+^ effects. 

**Tl(I).** The proximity of the crystal chemical radii of K^+^ and Tl^+^ allows the latter to use K^+^ transport systems to enter the cell [6,7,8]. Na^+^/K^+^-ATPase, the Na^+^-K^+^-2Cl^−^ cotransporter, and cellular membrane potassium channels are the main systems transporting Tl^+^ ions into cells [9,10,11,12]. Easy polarizing, compared to K^+^, allows Tl^+^ to form stronger chemical bonds with reactive electron donor groups of biological molecules (R-CO_2_^−^, R-OPO_3_^2−^, and R-SH). For this reason, compared to K^+^, Tl^+^ has a markedly greater affinity with cellular potassium channels and transporters [10,13]. These features underlie thallium toxicity in particular. The transport of K^+^ was inhibited by Tl^+^ in rat myocardial culture cells [14]. Tl^+^ penetrated the cytoplasm and damaged cellular and mitochondrial membranes in omnifarious body organs [9,15,16,17,18]. Thallium uncoupled oxidative phosphorylation in the mitochondria of mammalian cells [7,19,20,21]. The cytoplasmic concentration of Ca^2+^ and Na^+^ increased, while that of K^+^ fell due to the Tl^+^ toxic influence on rat hepatocytes and cardiomyocytes [22,23]. Unlike bivalent heavy metals, Tl^+^ does not have any noticeable effect on either mitochondrial respiratory enzymes (NADH-, succinate-, and malate-dehydrogenases) or mitochondrial thiol groups [9,10,24,25,26]. Here, it must be emphasized that the complexation constants of Tl^+^ with molecules with vicinal thiol groups turned out to be two orders of magnitude lower than those of ions of toxic heavy metals (Ag^+^, Hg^2+^, Cd^2+^, and Pb^2+^) [27].

**Hg(II).** It was previously shown that Hg^2+^ ions induced oxidative stress that manifested in a decrease in the reduced glutathione concentration and an increase in reactive oxygen species (ROS) or H_2_O_2_ production, lipid peroxidation, and state 4 respiration in experiments with energized rat kidney mitochondria (RKM) [28,29,30,31]. At the same time, there was the inner mitochondrial membrane potential (ΔΨ_mito_) decline, Ca^2+^ release, a respiration decrease in 3 and 3U_DNP_ (2,4-dinitrophenol (DNP)-uncoupled) states, an attenuation of atractyloside-insensitive ADP uptake, and a decrease in both Ca^2+^ accumulation and mitochondrial ATPase activity (of both basal and Mg^2+^-activated oligomycin-sensitive components). A Hg^2+^-induced respiration decrease was detected in experiments with RKM energized by substrates of the inner mitochondrial membrane complexes: CI and CII but not CIV [28]. The above toxic Hg^2+^ effects were accounted for by a reaction of Hg^2+^ with CI and CIII, and they could be additionally potentized by Ca^2+^ followed by the blocking of ruthenium red (RR) or ethylene glycol-bis(β-aminoethyl ether) N,N,N′,N′-tetraacetic acid (EGTA) [30]. Some of these Hg^2+^ effects were partly prevented by DTT but not cyclosporine A (CsA) or Mg^2+^. Additionally, Hg^2+^ activated proton ejection from the matrix of isolated beef heart mitochondria [32]. The H^+^ ejection and K^+^ uptake were more substantial in similar research with fluorescein mercuric acetate [32]. It is very interesting that the collapse of ΔΨ_mito_ and a decrease in both state 3 respiration and Ca^2+^ accumulation were found in kidney mitochondria isolated from mercury-poisoned rats [33]. 

Hg^2+^ (up to 5 μM) and MeHg (up to 1.5 μM) reduced ΔΨ_mito_ in sheep T cells [34]. Hg^2+^ significantly disturbed the intracellular structure of L929 mouse fibroblasts and inhibited mitochondrial dehydrogenase activity [35]. Experiments with methylmercury hydroxide-exposed rats showed that resistance to Hg^2+^-induced oxidative stress was supposed to be associated with increasing glutathione (GSH) synthesis in the kidney cortex rather than with an increase in the GSH level itself [36].

**Cd(II).** Cd^2+^ has previously been shown to increase the passive permeability of the inner membrane to univalent cations (H^+^, K^+^, Na^+^, and Li^+^), since the swelling of non-energized RLM and RKM increased in the series of Li^+^ < Na^+^ < K^+^ < NH_4_^+^ in experiments with chloride media [37,38,39]. Therewith, Cd^2+^ built up K^+^ transport in energized RLM and rat kidney mitochondria (RKM) [37,38,39,40,41,42]. Cd^2+^ induced mitochondrial swelling, increased state 4 respiration and inorganic phosphate (Pi) uptake, attenuated state 3 and 3U_DNP_ respiration, and induced ΔΨ_mito_ decline and Ca^2+^ efflux in RLM and rat heart mitochondria (RHM) energized by the substrates of mitochondrial complexes (CI, CII, and CIV) [37,38,40,42,43,44,45,46,47]. The short-term Cd^2+^-induced state 4 increase was eliminated by RR but not by NEM [45,48]. Cd^2+^-induced mitochondrial uncoupling (state 4_0_ increase) increases in the presence of Pi because Cd^2+^ stimulates the mitochondrial Pi uptake via the Pi/H^+^ symporter [45]. Similar potentiating Cd^2+^-induced effects to these states were found in a buffer with KCl replaced by NaCl but not LiCl [37]. A state 4 respiration increase with proton gradient discharge was found in experiments with RLM in the presence of low Cd^2+^; however, high Cd^2+^ induced a short-term increase in respiration, which was followed by a potent respiratory decrease due to the Cd^2+^ inhibition of CI and CII. However, even high Cd^2+^ concentrations (more 100 μM) did not inhibit state 4 respiration in RLM energized by a CIV substrate (ascorbate + N,N,N,N-tetramethyl-p-phenylenediamine (TMPD)) [38,43,44,49]. Cd^2+^ induced the efflux of protons into a sucrose medium containing 10 mM Me^+^ acetate in the series of Na^+^ < Li^+^ < K^+^ [41]. So, Cd^2+^ was concluded to inhibit CI–CIII but not CIV. DTT and RR prevented these Cd^2+^ effects. La^3+^ inhibited the Cd^2+^-induced swelling of succinate-energized RLM in a sucrose medium [41]. The chronic administration of low-dose CdCl_2_ to rats induced oxidative stress that resulted in a hepatic lipid peroxidation increase, hepatic glutathione depletion, and hepatic nuclear DNA damage in liver and brain mitochondria [50].

We showed earlier in the research on succinate-energized RLM [38,51] that the Cd^2+^-induced respiration decrease in 3 and 3U_DNP_ states and the state 4 respiration increase were inhibited by bivalent metal ions (Sr^2+^ and Mn^2+^) and RR (a blocker of the mitochondrial Ca^2+^ uniporter (MCU)) but markedly potentiated by Ca^2+^. This circumstance was mentioned subsequently by Belyaeva et al. [42,52] in similar experiments with both RR and other analogs of the MCU blocker, an analog of ruthenium red (Ru360). Sr^2+^, Mn^2+^, ruthenium red, and CsA prevented the Cd^2+^-induced swelling increase in succinate-energized RLM added to a buffer containing NH_4_NO_3_, but Ca^2+^ accelerated it. The Cd^2+^-potentiated swelling of the mitochondria in a K-acetate medium was increased in the presence of ruthenium red [37,38] or Ca^2+^ [52]. Ruthenium red prevented the Cd^2+^-induced decrease in ^137^Cs^+^ uptake and the swelling increase in a sucrose medium in experiments with succinate-energized RLM [37,38,52,53]. Injected before Cd^2+^, reagents (RR, DTT, and EGTA) visibly inhibited both the Cd^2+^-induced state 4_0_ increase and the state 3U_DNP_ decrease in rat liver and rat kidney mitochondria [42,51,54,55]. We previously suggested [51] that Cd^2+^ (like Ca^2+^, phenylarsineoxide, and other sulfhydryl reagents) could induce the opening of the mitochondrial permeability transition pore (MPTP) in the inner mitochondrial membrane. This assumption was confirmed in our subsequent experiments with a specific blocker of this pore (cyclosporine A, CsA), which significantly inhibited the Cd^2+^-induced swelling of RLM in various salt media [42,52]. 

The Cd^2+^ inhibition of the membrane-bound succinate dehydrogenase of beef heart was weakened by succinate and malonate [56]. Heavy metals (Hg^2+^, Cd^2+^, and Zn^2+^) with a high affinity for molecular SH groups attenuated Ca, Mg-ATPases’ activity and resulted in oxidative stress disturbing mitochondrial functions in the cells of different organisms [57]. Rod-selective apoptosis resulted in Ca^2+^ and/or Pb^2+^ elevation in rat photoreceptor cells [58]. Mitochondrial depolarization, an ATP synthesis decrease, swelling, release, and caspase-9 and caspase-3 activation were observed. These effects (Ca^2+^ and Pb^2+^) were additive and blocked completely by CsA [58,59].

**Pb(II).** Ruthenium red, N_3_^-^, and La^3+^ were previously shown to inhibit the uptake of Pb^2+^ by isolated rat kidney mitochondria [60]. Ca^2+^ uptake inhibition, ΔΨ_mito_ decline, a swelling and state 4_0_ increase, a state 3 decrease, and a fast release of accumulated Ca^2+^ were induced by Pb^2+^ in experiments with RKM, which were energized by CI–CII substrates [61,62,63]. RR prevented the Pb^2+^-induced ΔΨ_mito_ decline, state 3 respiration decrease, and Ca^2+^ release in RKM energized by succinate but not by NAD-dependent substrates [60,61]. In experiments with bovine heart mitochondria (BHM), energized by CI–CII substrates, Pb^2+^ inhibited Ca^2+^ uptake; resulted in mitochondrial swelling and ΔΨ_mito_ decline; increased the inner mitochondrial membrane (IMM) passive K^+^ permeability; and decreased K^+^ uptake and mitochondrial respiration in 4, 3, 3U_CCP_, and 3U_DNP_ states [64,65,66]. Some of these Pb^2+^ effects were blocked by Pi or chlorpromazine [66]. 

In vivo experiments with rats injected with ^203^Pb found appreciable lead quantities in the protein fractions but not in the lipid fractions of the inner membrane, the matrix, and the outer membrane of rat liver [67]. Research with rats exposed to Pb(NO_3_)_2_ found hepatocyte apoptosis with an activity decrease in the mitochondrial tricarboxylate carrier [68]. Pb^2+^ and Ca^2+^ competed for the plasma membrane transport systems and Ca^2+^-dependent effector mechanisms like calmodulin coupling several enzymes (phosphodiesterase and protein kinases) [69,70]. Pb^2+^ inhibited Ca^2+^ uptake in body organ cells and displaced Ca^2+^ and K^+^ from isolated mitochondria [69,71]. Pb^2+^ induced a cell viability decrease and a lipid peroxidation increase in experiments with cultured human fibroblasts [72]. 

**Al(III), Ga(III), and In(III).** Early research found that Al^3+^, some related metals (Sc^3+^, Ga^3+^, and In^3+^), and Be^2+^ stimulated carbonyl production and Fe^2+^-initiated lipid and protein oxidation in experiments in vitro with rat brain myelin and synaptic membranes [73]. Al^3+^ intoxication in vivo was manifested as an increase in lipid peroxidation and myelin fluidity in mouse brain [73]. Ga^3+^ and In^3+^ did not show any mitochondrial dehydrogenase inhibition or intracellular structure disruption in experiments with L929 mouse fibroblasts [35]. Experiments with rat brain myelin membranes showed that Fe^2+^-initiated lipid and protein oxidation decreased in the order of Sc^3+^, Y^3+^, La^3+^ > Al^3+^, Ga^3+^, In^3+^ > Be^2+^ [73].

**As(III).** AsO_2_^−^ markedly inhibited the Ca^2+^ uptake in RLM energized by a CI substrate (pyruvate) but not succinate [74]. The AsO_2_^−^-induced increase in state 4_0_ respiration in succinate-energized RLM was completely inhibited by butylhydroxytoluene (a free radical scavenger) or N,N′-dicyclohexylcarbodiimide (an F_1_F_O_-ATPase inhibitor) and partly inhibited by oligomycin [75]. State 3U respiration was completely inhibited in experiments with RLM energized by 2-oxoglutarate but not succinate in the presence of uncouplers (DNP and Cl-CCP) [76]. AsO_2_^−^ reacting with vicinal dithiols was found to enhance the H^+^-ATPase proton conductivity in the submitochondrial particles of beef heart mitochondria [77] and to inhibit beta-oxidation in isolated RLM [78]. These results indicate that arsenite significantly inhibits CI but not CII in these mitochondria. On the contrary, AsO_2_^−^ inhibited mitochondrial succinate dehydrogenase in mouse neuroblastoma N2a cells, but H_2_AsO_4_^−^ stimulated it [19]. Glutathione oxidation and the cross-linking of vicinal dithiols by arsenite or phenylarsine oxide (PAO) increased the probability of MPTP opening in RLM [79]. However, NEM eliminated this effect. However, Pi (H_2_PO_4_^−^) or arsenate (H_2_AsO_4_^−^) in Ca^2+^-loaded RLM induced MPTP opening, which was followed by an increase in H_2_O_2_ production, lipid peroxidation, and mitochondrial swelling [80].

**Cr(VI).** Cr_2_O_7_^2−^ resulted in lysosomal membrane rupture, reduced glutathione oxidation, increased ROS production and lipid peroxidation, and ΔΨ_mito_ decline in isolated rat hepatocytes [81]. Antioxidants, ROS scavengers, and Mn^2+^ prevented this Cr_2_O_7_^2−^-induced cytotoxicity. Mn^2+^ was supposed to prevent the formation of active Cr(IV) intermediates determining the toxic effects of Cr_2_O_7_^2−^ [81,82].

## 2. Modern Research on Heavy Metal Toxicity

**Ag(I). *Mitochondrial research.*** Low Ag^+^ showed a state 4 increase, but high Ag^+^ resulted in ΔΨ_mito_ decline and the complete inhibition of state 3 and state 4 respiration as well as an increase in the inner membrane proton and potassium permeability, ROS production, lipid peroxidation, and swelling in experiments with RLM energized by glutamate or succinate due to the inhibition of mitochondrial CI and CII but not CIV [83,84,85]. Only DTT (a thiol reagent) completely inhibited these Ag^+^ effects, while MPTP inhibitors (CsA and ADP), ethylenediaminetetraacetic acid (EDTA, a bivalent metal chelator), and RR (a mitochondrial Ca^2+^ uniporter blocker) had no effect [84,85]. S-15176 but not CsA inhibited Ag^+^-induced swelling, cytochrome c release, and MPTP opening in succinate-energized RLM [85,86]. Silver nanoparticles (AgNPs) induced MPTP opening and an increase in mitochondrial swelling, state 4 respiration, lipid peroxidation, and ROS production as well as ΔΨ_mito_ decline, a state 3 and Ca^2+^ retention decrease, and a negligible state 3U_FCCP_ decline in experiments with succinate-energized RLM [87,88]. The increase in AgNP-induced swelling was noticeably suppressed by DTT and ethylene glycol-bis(β-aminoethyl ether) N,N,N′,N′-tetraacetic acid (EGTA) but not by MPTP inhibitors (ADP and CsA), and the swelling was additionally accelerated in the presence of RR [87]. AgNP-induced mitochondrial swelling additionally increased in Ca^2+^-loaded RLM, which was followed by the inhibition of CsA [88]. 

**Ag(I). *Cell research.*** Ag^+^ induced mitochondrial apoptosis in human Chang liver cells, Ewing’s sarcoma A673 cells, A549 lung cancer cells, human airway smooth muscle, human breast SKBR3 cancer cells, histiocytic lymphoma U-937 cells, human promyelocytic HL-60 cells, human keratinocytes, CHO-K line cells, human bronchial epithelial BEAS-2B cells, human skin fibroblasts, and mouse AML12 hepatocytes [89,90,91,92,93,94,95,96,97,98,99,100,101,102]. The apoptosis manifested alongside decreased cell viability, caspase 3 and 7 activation, pro-apoptotic genes (*Bax* and *Bcl-2*) expression, increased metallothionein expression, ATP depletion, and DNA fragmentation. Oxidative stress was also present here as a decline in mitochondrial oxygen consumption, mitochondrial uncoupling, the inhibition of inner mitochondrial membrane complexes I and II (CI and CII), a decrease in reduced glutathione, the loss of ΔΨ_mito_, an increase in ROS production and lipid peroxidation, and cytochrome c release from the mitochondria as well as an increase in cytoplasmic and mitochondrial Ca^2+^. Some of these Ag^+^ effects were prevented by thiol reagents (DTT, N-acetylcysteine (NAC), and reduced glutathione). 

Silver nanoparticles (AgNPs) have found wide application in both biomedical and consumer products [89]. Silver nanoparticles (AgNPs) and Ag^+^ with a 1,10-phenanthroline complex induced mitochondrial apoptosis with caspase-3 and caspase-9 activation, decreased cell viability, and disrupted plasma and mitochondrial membrane integrity in experiments with hepatoblastoma HepG2 cells and A549 lung cancer cells [89,97,103]. AgNP-induced mitochondrial oxidative stress manifested as a decrease in reduced glutathione, mitochondrial uncoupling, ΔΨ_mito_ decline, a decrease in the antioxidant enzymes’ activity, ATP synthesis stoppage, an increase in membrane leakage, ROS production, lipid peroxidation, and NADPH oxidase activity in experiments with BRL-3A hepatocytes, human keratinocytes HaCaT cells, human peripheral blood mononuclear cells, and mouse AML12 hepatocytes [102,103,104,105,106]. Microscopic studies found that silver-nanoparticle-exposed cells showed an abnormal size, cellular shrinkage, and irregular shape acquisition [104]. The interplay between AgNP-induced mitochondrial fission and mitophagy defects was shown in in vivo and in vitro experiments with human hepatocellular carcinoma HepG2 cells [107]. The features of the AgNP reaction with cellular and mitochondrial enzymes and structures allow for the further use of these particles in different areas, such as anticancer and antimicrobial therapy; the textiles, water treatment, and cosmetics industries; and biomedical research as antimicrobial, antifungal, antiviral, anti-inflammatory, and anti-angiogenic agents [105,106,107,108]. 

The crystal chemical radii of Ag^+^ and K^+^ ions are close in magnitude; therefore, Ag^+^ can be transported through biological membranes via potassium channels. However, although Ag^+^ has the highest affinity for the thiol groups of biomolecules and cellular structures, it cannot be classified as the most highly toxic agent. The general explanation is the ability of Ag^+^ to react with Cl^−^ anions and metallothioneins binding heavy metal ions. So, these interactions markedly decrease the active concentration of Ag^+^ in the body’s internal environment. Silver nanoparticles are devoid of these shortcomings, and they are able to reach critical thiol groups. Recent comparative studies of AgNPs and Ag^+^ indicate this circumstance [109,110,111]. It was AgNPs that maximally reached and integrated into cellular structures, exerting a more substantial toxic effect on cells than Ag^+^. Understanding the factors determining the toxicity of AgNPs is very important for biomedical applications, particularly in cancer therapy. In this regard, modern studies of silver nanoparticles are very relevant in terms of the search for new anticancer and antimicrobial drugs. The Ag^+^ ions that overcome these barriers damage DNA, inhibit respiratory chain complexes, and induce MPTP opening, which are accompanied by the calcium overload of mitochondria. Together, Ag^+^ ions cause oxidative stress (Figure 1), which is accompanied by a decrease in ATP synthesis and the development of apoptotic processes (Figure 2) leading to cell death. Undoubtedly, an excessive accumulation of silver in the body can contribute to developing diseases associated with metabolic disorders and lead to the degradation of body systems and the development of oncology.

**Tl(I). *Mitochondrial research.*** The inner mitochondrial membrane turned out to be noticeably permeable to Tl^+^ ions [112,113]. We have shown that excess Tl^+^ ions are removed from mitochondria when using a K^+^/H^+^ exchanger, which is followed by the pumping out of incoming protons from the matrix [114]. In addition, Tl^+^ does not inhibit mitochondrial respiratory enzymes [9,10,24,25]. This is why Tl^+^ increased state 4 respiration and did not affect state 3 or 3U_DNP_ respiration in energized RLM [9,24,25,114,115,116,117]. As a similarly toxic heavy metal, Tl^+^ increased the passive permeability of the IMM to univalent cations (K^+^, H^+^, and Na^+^), and that was shown in experiments with non-energized RLM injected into a nitrate medium containing a mixture of nitrates of Tl^+^ and one of the univalent cations [114]. Energized RLM added to the nitrate medium showed an increase in swelling and state 4 respiration as well as some decline in state 3 and 3U_DNP_ respiration and the inner mitochondrial membrane potential (ΔΨ_mito_) [114]. 

It has been previously postulated that thallium(I) toxicity is due to the ability of Tl^+^ to easily penetrate the IMM and uncouple oxidative phosphorylation [24,112]. Tl^+^ ions transport into energized mitochondria using both the mitochondrial ATP-sensitive potassium channel (mitoK_ATP_) and the mitochondrial BK-type Ca^2+^-activated potassium channel (mitoK_Ca_) [118,119]. Massive mitochondrial swelling was found in in vivo experiments using thallium(I) salts [15,25,120]. One reason for this swelling may be MPTP opening in the inner mitochondrial membrane. We found that such phenomena as mitochondrial swelling, state 3 and 3U_DNP_ decrease, and ΔΨ_mito_ decline were markedly increased in experiments with energized calcium-loaded mitochondria [121,122]. These effects were completely eliminated or markedly attenuated by MPTP inhibitors (ADP, CsA, bongkrekic acid (BKA), and NEM), mitochondrial Ca^2+^ uniporter blockers (RR, Y^3+^, La^3+^, Sr^2+^, and Mn^2+^), or a Ca^2+^- chelator (EGTA) [115,121,123,124,125]. Tl^+^-induced MPTP opening is possible only in the case of calcium-loaded mitochondria, while such loading is not required in similar experiments with heavy bivalent metals. The inhibition of mitoK_ATP_ and mitoK_Ca_ decreased the matrix calcium retention and accelerated the Tl^+^-induced MPTP opening in calcium-loaded rat liver mitochondria [126]. The substrate specificity effect [127] was found in succinate-energized mitochondria, which had more resistance to calcium loading than those energized by CI substrates [121]. Tl^+^-induced MPTP opening is dependent on the adenine nucleotide translocase (ANT) conformation [124]. The relationship between the ANT conformation and MPTP opening in the inner membrane was substantiated previously in detail [128,129]. The pore-opening phenomena were more noticeable in fixing the ANT c-conformation induced by thiol reagents (phenylarsine oxide (PAO), 4,4′-diisothiocyanostilbene-2,2′-disulfonate (DIDS), and high NEM), thiol oxidizers (tert-butyl hydroperoxide (*t*BHP), diamide (Diam)), and carboxyatractyloside (CAT) [124,130]. On the contrary, the manifestation of these phenomena noticeably decreased or completely inhibited the stabilization of the ANT m-conformation induced by reagents (ADP, low NEM, BKA, and eosin-5-maleimide (EMA)) and low mersalyl [9,124,131]. Additionally, Tl^+^-induced MPTP opening was recently found to be dependent on the activity of cysteine and lysine residues in the inner membrane proteins [131,132]. 

**Tl(I). *Cell research.*** Apoptosis, membrane integrity disruption, MPTP opening, Na^+^/K^+^-ATPase inhibition, glutathione depletion, ΔΨ_mito_ decline, cell viability loss, cytochrome c release, membrane fluidity rise, ATP depletion from mitochondria, and an increase in lipid peroxidation, ROS production, and H_2_O_2_ synthesis were induced by Tl^+^ in experiments with isolated rat hepatocytes, isolated synaptosomal mitochondrial fractions of P2, PC12, and HN9.10e neuronal cells, C6 glioma and Jurkat T cells [9,115,133,134,135,136,137,138,139,140,141,142]. Tl^+^-induced apoptosis in PC12 cells was later shown to be associated with the activation of the MAPK-dependent cascade of protein kinases: extracellular signal-regulated kinase (ERK)1/2, c-jun N-terminal kinase (JNK), p38, and p53 of the *Bcl-2* family [138]. Experiments with TlCl-treated rat hepatocytes found increased cytoplasmic concentrations of Ca^2+^ and Na^+^ but decreased K^+^ concentrations [22]. MPTP inhibitors (CsA and carnitine) visibly reduced the cytotoxic effects of Tl^+^ [136,137]. Thallium(I)-induced metabolic disturbances are situations in which Tl^+^ inhibits Mg^2+^-regulated and K^+^-dependent pyruvate kinases, disrupts glycolysis processes, and interferes with the Krebs cycle [7,9,10,12,13]. 

However, it is very likely that the Tl^+^-induced decrease in reduced glutathione in the matrix was due to our hypothesized reversible oxidation of Tl^+^ to Tl^3+^ near the centers of the generation of ROS in the respiratory chain [9]. It should be noted that thallium toxicity to humans turned out to be higher than the toxicity of mercury, lead, cadmium, copper, and zinc [21,143]. 

Tl^+^ and K^+^ cations (as well as Ag^+^) have similar crystal chemical radii. For this reason, Tl^+^ ions enter cells through potassium channels. However, Tl^+^ (unlike Ag^+^ and other heavy metals) has a minimal affinity for thiol groups. For this reason, Tl^+^ really does not bind to intracellular metallothioneins and quickly penetrates cells, disrupting potassium-dependent cell processes. This distinguishing Tl feature is especially true of the toxic effects of Tl^+^ ions on neurons, cardiomyocytes, and kidney cells. With a more prolonged exposure to the body, Tl^+^ reduces the concentration of reduced glutathione while increasing ROS production in cells. Oxidative stress (Figure 1) and apoptotic processes (Figure 2) develop due to damage to cellular and mitochondrial membranes, decreased ATP synthesis, and the subsequent induced MPTP opening and calcium overload of cells and mitochondria. Such features of the chemical behavior of Tl^+^ do not allow this metal to be rapidly removed from the body. Future research should focus on finding effective Tl^+^ binding reagents for the subsequent elimination of this metal from the body to treat thallium-induced damage to the nervous system, kidneys, and hairline [9]. 

**Hg(II). *Mitochondrial research.*** Hg^2+^ was shown to induce oxidative stress in experiments with RLM energized by CI and CII substrates [42,144,145,146]. There was a state 3 and 3U_DNP_ respiration decrease, ΔΨ_mito_ decline, mitochondrial swelling, a state 4 respiration increase, a mitochondrial membrane fluidity increase, cytochrome c release, and MPTP-induced mitochondrial ATP depletion. So, Hg^2+^ to Cys binding mediates multiple Hg^2+^ toxic impacts, especially the inhibition of enzymes and other proteins containing free Cys residues causing oxidative stress [147]. These unfavorable effects were prevented by MPTP inhibitors (ADP, CsA, Mg^2+^, and BKA) and Ru360 (an MCU blocker). F_1_F_O_-ATPase activity was promoted by micromolar Hg^2+^, and respiration in state 4 was inhibited in swine heart mitochondria energized by NADH but not succinate [148]. Hg^2+^ and MeHg significantly declined mitochondrial viability and increased the H_2_O_2_ and ROS production, lipid peroxidation, and glutathione oxidation in mouse brain mitochondria [149]. Quercetin and catalase prevented these effects of Hg^2+^. Hg^2+^ resulted in swelling, a Ca^2+^ efflux, and an ROS production increase in experiments with RKM [150]. The calcium load of these mitochondria additionally increased swelling and induced ΔΨ_mito_ decline. Tamoxifen and CsA inhibited these in vitro effects of Hg^2+^. Ca^2+^ uptake and Ca^2+^-induced ΔΨ_mito_ decline were inhibited in mitochondria isolated from the kidneys of rats injected with mercury chloride plus tamoxifen [150]. Tamoxifen was concluded to be an inhibitor of Hg^2+^-induced MPTP [150]. 

**Hg(II). *Cell research.*** MeHg similarly to Hg^2+^ induces mitochondrial apoptosis and oxidative stress, mediating an increase in ROS production, a decrease in cell antioxidant defense, and cytochrome c release from mitochondria [151,152]. Hg^2+^ induced apoptosis with caspase-3 activation, a cell viability decrease, mitochondrial integrity decline, and intracellular ATP depletion. There was oxidative stress manifested as a decline in ΔΨ_mito_, cytochrome c release, an increase in ROS generation, an increase in lipid peroxidation, a decrease in ATP and reduced glutathione, and a decrease in thioredoxin reductase and glutathione peroxidase activity in human hepatoma HepG2 cells, human neuroblastoma SH-SY5Y cells, human gingival fibroblasts, hamster pancreatic HIT-T15 β-cells, normal rat kidney cells, rat ascites hepatoma AS-30D cells, and human T cells and leukocytes [153,154,155,156,157,158,159,160,161,162]. N-acetylcysteine reversed some of these effects of Hg^2+^. Experiments with Hg^2+^- and MeHg-treated fibroblast zebrafish ZF4 cells found a decrease in ATP production and mitochondrial respiration in the basal, 3, and 3U_FCCP_ states [163]. However, the proton leak and non-mitochondrial respiration were not changed there [163]. Hg^2+^ induced a cell viability decrease, ΔΨ_mito_ decline, and an ROS production increase in experiments with five Tetrahymena species; however, more ingested MeHg disrupted the membrane integrity [164]. 

The Hg^2+^- and MeHg-induced apoptosis in HepG2 cells can be due to Hg^2+^, and MeHg inhibited the respiration, increased the permeability, and blocked the essential thiol proteins of mitochondrial membranes [165]. The inner membrane protein thiols were dose-dependently decreased by Hg^2+^ in swine heart mitochondria [148]. However, these Hg^2+^-induced effects were reversed by DTT due to the latter’s interaction with the Cys residues of respiratory complexes [148]. A Fenton-type reaction may be the cause of the Hg^2+^-induced oxidative stress and ROS production in NRK-52E cells due to the binding of Hg^2+^ to intracellular thiols (proteins or glutathione) [166]. Hg^2+^ increased free Ca^2+^ levels in canine kidney cells due to both extracellular Ca^2+^ influx (protein kinase C-regulated) and the phospholipase C-activated Ca^2+^ release from endoplasmic reticulum (ER) [167]. MeHg-induced neurotoxicity in primary rat cortical neurons stimulated Ca^2+^ release, increasing from the ER simultaneously with MPTP opening and ΔΨ_mito_ decline [168]. 

Hg^2+^ penetrates cells and mitochondria via calcium-transporting mechanisms. Hg^2+^ ions induce oxidative stress (Figure 1) and apoptotic processes (Figure 2) in cells and mitochondria, having a noticeable affinity for thiol groups. This stress manifests in a marked decrease in CI and CII activities, a decline in ΔΨ_mito_, the induction of MPTP opening, and a decrease in ATP and GSH synthesis. At the same time, the calcium overload of cells and mitochondria sharply increases, ROS production increases, and the reduced GSH concentration decreases. At higher levels, Hg^2+^ damages renal nephrons. The main toxic manifestations in this case can be considered the induction of MPTP opening and the blocking of critical thiol groups by Hg^2+^ ions because the MPTP inhibitors (ADP, CsA, and tamoxifen) and reducers of these groups (DTT and NAC), along with calcium transport blockers (RR and Ru360), eliminated the toxic effects of Hg^2+^. 

**Cd(II). *Mitochondrial research.*** Studies of the cadmium effects on various characteristics of isolated mitochondria were conducted. Cd^2+^ in succinate-energized RLM induced ΔΨ_mito_ decline; iron mobilization; a decrease in state 3 and 3U_DNP_ respiration, ATP content, potential dependent ^137^Cs^+^ uptake, and protein thiol levels; and an increase in state 4_0_ respiration, IMM proton permeability, and H_2_O_2_ production; but Cd^2+^ slowly affected RLM succinate dehydrogenase activity [48,54,144,146,169]. Cd^2+^ (similarly Hg^2+^) increased the inner membrane passive permeability to H^+^ and K^+^ ions and showed ΔΨ_mito_ collapse, a decrease in 3U_DNP_ respiration, and an increase in K^+^ mitochondrial uptake and state 4_0_ respiration in RHM energized by CI and CII substrates [42,46,47]. Cd^2+^ resulted in respiration inhibition in 3 and 3U_FCCP_ states, ΔΨ_mito_ decline, cytochrome c release, Ca^2+^ uptake inhibition, and a Ca^2+^ release and H_2_O_2_ generation increase in isolated rat kidney mitochondria [39,55,170]. DTT and CsA prevented the Cd^2+^-induced Ca^2+^ efflux and ΔΨ_mito_ decline in energized rat kidney mitochondria [171]. Quinine inhibited the contraction of succinate-energized RKM preswollen in a KCl medium containing Cd^2+^ [39]. 

After reacting with the outer and inner thiol groups of the inner membrane, Cd^2+^ and Hg^2+^ then penetrate into mitochondria and cause the release of cytochrome c, the opening of MPTP, and the inhibition of CI and CII [42]. After penetrating into the matrix through the mitochondrial Ca^2+^ uniporter, Cd^2+^ was assumed to induce mitochondria osmotic swelling and the cytochrome c release due to the activation of aquaporin H_2_O channels because mitoplast swelling was blocked by Ag^+^ (a potent aquaporin blocker) [172]. The Cd^2+^-induced swelling, ΔΨ_mito_ decline, state 4_0_ increase, IMM fluidity increase, and lipid peroxidation were decreased in the presence of MPTP inhibitors (ADP, CsA, Mg^2+^, DTT, NEM, RR, and Ru360) in experiments with succinate-energized RLM [39,44,48,146,173]. Moreover, the maximum effect was achieved with the simultaneous presence of some of these inhibitors in salt media containing NH_4_NO_3_, KCl, or sucrose [44]. Atractyloside prevented the contraction of succinate-energized RLM preswollen in a NH_4_NO_3_ medium containing 15 µM Cd^2+^, oligomycin, and rotenone [44]. CsA partly inhibited the Cd^2+^-induced state 4_0_ increase and state 3U_DNP_ decrease in succinate-energized RLM [44]. Thus, Cd^2+^-induced mitochondrial dysfunction appears to be associated with the opening of ADP- and CsA-dependent MPT pores in the inner membrane [44]. Cd^2+^ resulted in MPTP opening, ΔΨ_mito_ decline, cytochrome c release, decreased state 3 respiration, and increased state 4_0_ respiration. These Cd^2+^ effects were completely suppressed by Bcl-xL, RR, and BKA but not by CsA in experiments with succinate-energized mouse liver mitochondria [174]. This Cd^2+^-induced MPTP opening was supposed to be the reason for the Cd^2+^ interaction with ANT thiol groups [174].

We have recently been shown that Mg^2+^ with the ADP inhibition of the state 4_0_ increase, induced by high Cd^2+^ with RR, was not changed in experiments with succinate-energized RLM added to a K-acetate medium containing both a mitoK_ATP_ inducer (diazoxide) and the channel inhibitors (5-hydroxydecanoate (5-HD) and glibenclamide) [146]. However, the Cd^2+^-induced swelling of RLM-energized glutamate with malate increased in a KCl medium containing the mitoK_ATP_ inhibitor ATP. This effect was slightly prevented by diazoxide and, in contrast, was markedly potentiated by 5-HD [146]. So, one can conclude that the above inhibition of the Cd^2+^-induced state 4_0_ increase is probably due to the mitoK^+^ uniporter blocking by ATP with Mg^2+^ but not the MPTP opening. On the contrary, the Cd^2+^-induced swelling increase in the KCl medium containing mitoK_ATP_ inhibitors (ATP and 5-HD) is probably due to the greater MPTP opening when the mitoK_ATP_ channels are closed. This result is in good agreement with the previous assumption that the probability of the MPT pore opening in the inner membrane increases when the mitoK_ATP_ channels are closed [175,176,177].

Research on energized RLM found that the state 4_0_ respiration increase and the state 3U_DNP_ respiration decrease induced by Cd^2+^ with Ca^2+^ were visibly eliminated in the presence of ADP with CsA [54,178]. However, Cd^2+^-induced state 4_0_ respiration increased even more after the administration of DTT but not EGTA, which was injected after high Cd^2+^ [54,144,178]. Ca^2+^ with Pi or NEM alone markedly increased the Cd^2+^-induced passive proton permeability of the inner membrane. In this case, preswollen RLM (after the succinate addition) showed some contraction in a NH_4_NO_3_ medium containing low NEM (10–100 μM) or Pi alone but not Ca^2+^ with/without Pi or high NEM (1 mM). This contraction became much more potent after the DTT addition to the medium containing NEM with rotenone, and the succinate-energized mitochondria visibly contracted (like the control Cd^2+^-free experiments), even in the presence of 1 mM NEM (except for in the experiments where Cd^2+^, Ca^2+^, and Pi were all present) [44,54,178]. This research concluded that Cd^2+^-induced mitochondrial dysfunction is due to the fact that this metal acts as a thiol and Me^2+^ binding site reagent [54]. CsA inhibited an additional Cd^2+^-induced increase in K^+^ transport in succinate-energized RLM in the presence of ruthenium red [178]. Experiments with cadmium metallothionein in vivo and Cd^2+^ in vitro found renal cortical mitochondria swelling and respiratory function decline (a decrease in state 3 and the respiratory control ratio (RCR)) that indicates both electron transfer and oxidative phosphorylation inhibition [179].

**Cd(II). *Cell research.*** Cd^2+^ transports into cells by using ion channels, special carriers, Ca^2+^/calmodlin-dependent protein kinase II (CaMK-II), and ATP-hydrolyzing pumps, whereas Cd^2+^–protein complexes penetrate the membrane through receptor-mediated endocytosis of the cadmium–metallothionein complexes but not N- and L-type Ca^2+^ channels, which are closed at the resting membrane potential [180,181,182]. In addition, the mitochondrial Ca^2+^ uniporter is involved in the Cd^2+^ transport into mitochondria [180]. 

Being a physiological secondary messenger, Ca^2+^ and ROS control Ca^2+^-dependent and redox-sensitive molecular processes that determine cell function and fate. Ca^2+^ and ROS are known to activate cell death effectors (caspases, ceramides, calpains, and p38), to irreversibly damage mitochondria and the endoplasmic reticulum, and to modulate biosynthesis metallothioneins and *Bcl-2* proteins [183]. It is the Ca^2+^ and ROS increase that is directly affected by the Cd^2+^ impact in experiments in vitro and in vivo [183]. Cd^2+^ produced an increase in the cytoplasm Ca^2+^ concentration in canine kidney cells, rat primary astrocytes, rat primary cerebral cortical neurons, yeast cells, mouse skin fibroblasts, and rat hepatocytes [184,185,186,187,188,189]. Cd^2+^ induced the release of the stored Ca^2+^ from the endoplasmic reticulum and mitochondria in canine kidney cells and NIH 3T3 cells [184,190]. Cd^2+^ decreased agonist-induced ER calcium signals and sarcoplasmic–ER calcium ATPases activity in NIH 3T3 cells [190]. 

Caspases (cysteine proteases), ceramides (cell membrane lipids), and calpains (Ca^2+^-dependent proteases) are known to play an important role in Cd^2+^-induced apoptosis in rat kidney proximal tubule cells and to be related directly to Cd^2+^-induced nephrotoxicity in vivo [191]. Cd^2+^ induced apoptosis involving the activation of caspase-9, caspase-3 and mitogen-activated protein kinases (ERK, JNK, p53, and p38) as well as DNA fragmentation in rat C6 glioma cells, HEK293 cells, Vero cells, rat ascites hepatoma AS-30D cells, human lung cancer A549 cells, hamster ovary CHO-9 cells, PC12 cells, rat and mouse cochlea HEI-OC1 cells, rat primary astrocytes, rat primary cerebral cortical neurons, U-937 human promonocytic cells, mouse skin fibroblasts, NIH 3T3 cells, yeast cells, rat kidney proximal tubule cells, brain newborn rat oligodendrocytes, rat hepatocytes, mouse skin epidermal JB6 Cl41 cells, AS-30D ascites hepatoma cells, and human hepatocellular carcinoma HepG2 cells [146,158,159,161,162,172,185,186,187,189,190,192,193,194,195,196,197,198,199,200,201,202,203,204,205,206,207]. This Cd^2+^-induced apoptosis was accompanied by oxidative stress and mitochondrial injury, which manifested as a cell viability reduction, CI–CIII dysfunction, ATP depletion, ΔΨ_mito_ decline, matrix Ca^2+^ load, a reduced glutathione decrease, cytochrome c release, an ROS and H_2_O_2_ production increase, and lipid peroxidation as well as mitochondrial damage and swelling. Cd^2+^-induced oxidative stress is affected by this metal interaction with sulfhydryl groups of mitochondrial critical molecules [208]. Na^+^/K^+^-ATPase, Ca^2+^/Mg^2+^-ATPase, and other transport proteins can be damaged by the Cd^2+^-induced ROS increase in rat kidney proximal tubule cells and rat primary cerebral cortical neurons [189,198]. Cd^2+^-induced actin disruption and the activation of Ca^2+^/calmodulin-dependent protein kinase II preceded the acceleration of apoptotic mouse and rat mesangial cell death involving actin-dependent mitochondrial structure changes, ROS production, and p38 activation [209,210]. Experiments with rats exposed to Cd acetate found Cd^2+^-induced rat liver oxidative stress manifesting as decreased superoxide dismutase (SOD) activity and GSH content as well as increased catalase activity and thiobarbituric acid content [211].

Cd^2+^ in normal human lung MRC-5 fibroblasts caused caspase-independent apoptosis mediated by ROS scavengers (NAC and mannitol) to indicate the crucial role of ROS in apoptogenic Cd-activated apoptosis. The Cd^2+^-induced apoptosis was partially or completely abrogated by CI and CV inhibitors (rotenone and oligomycin A) or MPTP inhibitors (CsA and aristolochic acid) [212]. 

Cd^2+^ and Hg^2+^ were known to induce metallothionein synthesis and alter zinc and copper homeostasis in the liver and kidney tissue of male albino mice [213]. EGTA and polyphenolic compounds (ellagic acid and quercetin) effectively reduced the Cd^2+^ toxic effects on rat primary astrocytes and goat sperm [185,214]. The Cd^2+^-induced Ca^2+^ release in renal tubular cells was inhibited by thapsigargin (an ER Ca^2+^pump inhibitor) and carbonyl cyanide m-chlorophenyl hydrazone (CCCP, a mitochondrial uncoupler) [184]. Mn^2+^ prevented the Cd^2+^-induced calcium increase in cultured cerebellar granule neurons and dose-dependent death rate [215]. Zn^2+^ inhibited the intracellular Cd^2+^ bioaccumulation in cultured rat cortical neurons and RAW 264.7 macrophages [216,217]. Zn^2+^ prevented Cd^2+^-induced necrosis and apoptosis in PC12 cells, bovine kidney epithelial cells, and rabbit proximal tubule RP1 cells [218,219,220]. Suppressed mitochondrial apoptosis and a decrease in Cd^2+^-induced ROS production and cytochrome c release, as well as the downregulation of the Bax proapoptotic protein and a lowering of caspase 9 expression, were observed. The Cd^2+^ effects on rat ascites hepatoma AS-30D cells and PC12 cells were partly attenuated by MPTP inhibitors (CsA and BKA), respiratory CIII inhibitors (stigmatellin and antimycin A), and antioxidants (N-acetylcysteine and butylhydroxytoluene) [159,204]. Ru360 blocked Cd^2+^-induced apoptosis in rat kidney proximal tubule cells, but MPTP inhibitors (CsA and BKA) had no effect [172,198]. Reduced glutathione and catalase markedly prevented Cd^2+^-induced DNA fragmentation in rat C6 glioma cells, suggesting the participation of H_2_O_2_ production in these apoptotic processes [193]. Cd^2+^ cytotoxicity was regulated by intracellular glutathione levels in newborn rat brain oligodendrocytes and rat kidney proximal tubule cells [197,199]. Antioxidants (N-acetyl-l-cysteine and ebselen) noticeably decreased Cd^2+^ toxicity on rat and mouse cochlea HEI-OC1 cells [196]. Glutathione S-transferase P1 overexpression greatly reduced the Cd^2+^ effects on HEK293 cells [194]. Apoptosis inhibitors (cycloheximide and dactinomycin), a caspase 3 inhibitor, MPTP inhibitors (CsA, carnitine, and trifluoperazine), ROS scavengers (dimethyl sulfoxide (DMSO), mannitol, catalase, and SOD), a Na^+^/H^+^ exchanger inhibitor (5-(N,N-dimethyl)-amiloride), and 1,2-bis(o-aminophenoxy)ethane-N,N,N′,N′-tetraacetic acid (BAPTA, a specific intracellular Ca^2+^ chelator) prevented Cd^2+^-induced cytotoxicity and ΔΨ_mito_ decline in rat hepatocytes and rat cerebral cortical neurons [188,189,200,202]. NO prevented Cd^2+^-induced apoptosis, caspase-3 activation, and lipid peroxidation in rat anterior pituitary cells [221]. 

Cd^2+^ actively interacts with the IMM thiol groups of cells and mitochondria. The result of this interaction is the induction of apoptosis (Figure 2) and oxidative stress (Figure 1) manifesting in increased ion transport through IMM, the inhibition of mitochondrial respiratory complexes (CI–CIII), a decrease in ATP synthesis, the calcium overload of mitochondria, an increase in ROS production, a decline in ΔΨ_mito_, and the induction of the MPTP opening. At a higher body level, Cd^2+^ damages target organ cells (nervous tissue and liver). Considering the closeness of Cd^2+^ and Ca^2+^ radii, it should be noted that the toxic effects of Cd^2+^ increase significantly under conditions of the calcium overload of cells and mitochondria. The above Cd^2+^ effects, regardless of the presence of calcium, were markedly attenuated in the presence of calcium transport blockers and competitors (RR, Ru360, Sr^2+^, and Mn^2+^), MPTP inhibitors (CsA and BKA), and thiol-reducing agents (NAC and DTT). Therefore, the main three toxic Cd^2+^ manifestations leading to oxidative stress and apoptosis can be considered the Cd^2+^-induced calcium overload of mitochondria, the blocking of critical thiol groups, and the opening of MPTP. Cadmium toxicity processes can involve mitochondrial s-glutathionylation. Cd^2+^ disturbed the redox balance of mitochondria due to an ROS production increase, which enhanced the S-glutathionylation of mitofusin 2 and damaged the membranes binding ER with subsequent neuronal necroptosis [222]. 

**Pb(II). *Mitochondrial research.*** Experiments with mitochondria showed that Pb^2+^-induced mitochondrial oxidative stress manifested in the inhibition of CII–CIV as well as in an increase in ROS production and lipid peroxidation, ATP consumption, and glutathione oxidation [223,224]. In parallel, Pb^2+^ decreased the activity of mitochondrial antioxidant system enzymes (GPX, SOD, catalase, and GSH) [224]. At the same time, Pb^2+^ induced MPTP opening that manifested in RLM swelling, a state 3 respiration decrease, ΔΨ_mito_ decline, an IMM fluidity decrease, proton influx in the matrix, and K^+^ and cytochrome c release. The Pb^2+^-induced swelling visibly declined in the presence of known MPTP inhibitors (CsA and ADP), chelators (EGTA and EDTA), and a mitochondrial Ca^2+^ uniporter inhibitor (RR) [71,223,224]. 

**Pb(II). *Cell research.*** Pb^2+^ induced apoptosis, including the activation of caspase-3 and the expression of the apoptotic-inducing factors (*Bax*, *Bcl2*, and *Bcl-2*), MPTP opening, and oxidative stress that manifested as mitochondrial function oxidative damage, cytochrome c release, and a decrease in cell viability, reduced glutathione, and ATP production as well as an ROS production increase, ΔΨ_mito_ decline, Ca^2+^ release from the matrix, and a decline in the mitochondrial Ca^2+^ uniporter expression in human neuroblastoma SH-SY5Y cells, rat proximal tubular cells, Siberian tiger fibroblasts, rat insulinoma β-cells, PC12 cells, and adult rat hepatic stem cells [225,226,227,228,229,230,231,232]. These effects of Pb^2+^ were eliminated by MPTP inhibitors (CsA, DIDS, and BKA), which may indicate the participation of Cyp-D and ANT in Pb^2+^-induced MPTP opening. It was concluded that the mitochondrial Ca^2+^ influx regulation in neurons is mediated by the Pb^2+^-induced oxidative stress response.

The mechanism of Pb^2+^ neurotoxicity is associated with this cation ability to replace or compete with important biogenic cations (Ca^2+^, Fe^2+^, and Zn^2+^) [233]. Pb^2+^-induced oxidative stress manifested in mitochondrial uncoupling; the inhibition of CI and CIII; a decrease in the matrix reduced glutathione; ΔΨ_mito_; mitochondrial O_2_ consumption; an increase in the IMM ion permeability, ROS production, and lipid peroxidation; and a decrease in cytoplasm ATP due to an energy consumption imbalance [233]. All these toxic Pb^2+^ effects led to the displacement of Ca^2+^ from intracellular compartments, the opening of MPTP, and the induction of apoptotic and necrotic changes in neurons and cells [233]. Pb^2+^ induced a reduction in glucose-stimulated insulin secretion in experiments with rat insulinoma β-cells [230]. 

The toxic actions of Cd^2+^, Hg^2+^, and Pb^2+^ were studied in vitro on pancreatic β-cells from CD-1 mice [234]. These metals resulted in oxidative stress manifesting as a decrease in ΔΨ_mito_, ATP production, CI–CIV activity, and oxygen consumption rates with a parallel increase in cell lactate production, mitochondrial succinate-supported swelling, and IMM permeability to K^+^ and H^+^ ions as well as an increase in membrane fluidity and a decrease in saturated/unsaturated fatty acid ratios [234]. Pb^2+^ damaged liver mitochondrial and nuclei structures and activated mitochondrial apoptosis through caspase-3 and caspase-9 signaling pathways in experiments with Pb-poisoned chickens [235]. The independence of the toxic mechanisms of Cd and Pb was proven in experiments with rats exposed to Cd acetate [211]. Pb^2+^ alone resulted in a decrease in liver superoxide dismutase (SOD) activity; however, catalase activity and glutathione (GSH) or thiobarbituric acid content were unchanged. Cd^2+^ with/without Pb^2+^ resulted in rat liver oxidative stress because a decrease in SOD activity and GSH content and an increase in catalase activity and thiobarbituric acid content were found. 

A distinctive feature of Pb^2+^ ions is their ability to compete with Ca^2+^ for calcium transport systems of the plasma membrane. However, by entering the mitochondria, Pb^2+^ inhibits Ca^2+^ uptake and displaces Ca^2+^ and K^+^ from the matrix. The main targets of Pb^2+^ are the cells of the nervous system, pancreas, and liver. Given the ability of lead to displace calcium from cells, osteoblasts may be another reason for Pb^2+^ toxic effects leading to bone demineralization. After penetrating mitochondria, Pb^2+^ induces oxidative stress (Figure 1) and apoptotic processes (Figure 2), interacting with thiol groups and displacing essential biogenic cations (Ca^2+^, Fe^2+^, and Zn^2+^) from mitochondrial structures. Such oxidative stress is characterized by decreased CII–CIV activity, MPTP opening, ΔΨ_mito_ decline, decreased ATP consumption, increased ROS production and lipid peroxidation, and glutathione oxidation, reducing the activity of mitochondrial antioxidant system enzymes. That is why Pb^2+^ toxicity visibly declines in the presence of known MPTP inhibitors (ADP, CsA, BKA, and DIDS), chelators (EGTA and EDTA), antioxidants (catalase), and a mitochondrial Ca^2+^ uniporter inhibitor (RR). Since lead enters the environment due to various technological processes, it has become essential to develop ways to protect the body from the toxic manifestations of lead compounds and remove them from the body. Pb^2+^ can accumulate in astrocytes and have a pronounced poisonous effect, disrupting glutamate metabolism [236]. Additionally, the activation of S-glutathionylation processes was found because of the enhanced expression of glutathione–protein complexes after Pb treatment. 

The accumulation of heavy metals (Hg, Cd, Pb, Cr, U, and As) in living organisms leads to toxic damage to different organs and tissues. These heavy metals disturb various biological functions, including proliferation, differentiation, damage repair, and apoptosis induction [237,238,239]. At the same time, the mechanisms of toxicity induction often have common patterns directly related to the occurrence of mitochondrial oxidative stress. An ROS production increase, antioxidant defense reduction, enzyme inactivation, MPTP opening, calcium overload, and ΔΨ_mito_ decline belong to these patterns. The toxic effects of these metals are multilevel and may play a role in the development of cancer, autoimmune diseases, diabetes, toxicogenomic disorders, Parkinson’s disease, and other metabolic disorders.

**Al(III), Ga(III), and In(III). *Mitochondrial research.*** Al^3+^ decreases the Ca^2+^ uptake in ER, induces the release of Ca^2+^ from the matrix, and potently inhibits Ca^2+^-ATPase activity in RLM that can result in intracellular calcium overload [240]. Aluminum nanoparticles (AlNPs), more so than Al^3+^, induced oxidative stress that manifested as an increase in mitochondrial swelling, ROS production, and lipid peroxidation, as well as CIII inhibition, ΔΨ_mito_ decline, and cytochrome c release, in experiments with isolated rat brain mitochondria [241]. Al^3+^ plus tyramine induced MPTP opening in Ca^2+^-loaded RLM that manifested as mitochondrial swelling, an ROS and H_2_O_2_ production increase, mitochondria morphological alterations, and the oxidation of pyridine nucleotides and mitochondrial thiols [242]. These effects were completely eliminated by MPTP inhibitors (CsA, DTT, and NEM) and partly eliminated by catalase. The opening of this pore requires the simultaneous oxidation of thiol groups on both sides of the inner mitochondrial membrane [242].

In^3+^ in experiments with succinate-energized RLM induced mitochondrial swelling, ΔΨ_mito_ decline, ROS production, a proton permeability increase, and electron transition inhibition [243]. In^3+^-induced MPTP was due to the inhibition of the inner membrane proton channels and the stimulation of mitochondrial oxidative stress, resulting in a 15–20% decrease in mitochondrial respiration in states 4, 3, and 3U_DNP_ [243]. The In^3+^-induced swelling, ΔΨ_mito_ decline, and proton permeability increase were markedly eliminated by MPTP inhibitors (ADP and CsA), chelating agents (EGTA and EDTA), and ruthenium red but not by the thiol reagent DTT. 

**Al(III), Ga(III), and In(III). *Cell research.*** Al(III) exists in water as a hydroxocomplex Al(H_2_O)_6_^3+^ that reacts with superoxide radical O_2_^•−^ to form semireduced radical AlO_2_(H_2_O)_5_^2+•^ that depletes mitochondrial Fe followed by the generation of H_2_O_2_, O_2_^•−^, and OH^•^ [244]. Both Al^3+^ and aluminum compounds target body tissues resulting in bone mineralization impairment, hemoglobin synthesis impairment, and cardiovascular and neurological disorders [244]. Al^3+^ ions additionally increased glutamate-induced intracellular Ca^2+^ overload in rat hippocampal organotypic cultures [245]. Al^3+^ resulted in an intracellular Ca^2+^ increase followed by the inhibition of protein kinase B phosphorylation, resulting in cell morphology impairment in PC12 cells [246]. 

Al^3+^-induced apoptosis accompanied by the activation of caspases-9, -8, and -3 and oxidative stress manifested as a cell viability decrease, cytochrome c release, and ΔΨ_mito_ decline in experiments with rat osteoblasts [247,248]. Al^3+^ induced an ROS generation increase and mitochondrial damage, resulting in a decrease in SOD and catalase activities and ATP synthesis in hippocampal neuronal cells and human HepG2 cells [249,250,251]. These effects were attenuated by chlorogenic acid [249]. Al^3+^-induced oxidative stress showed toxic effects that manifested as cell viability collapse, ΔΨ_mito_ decline, an increase in ROS production, and a decrease in oxygen consumption and mitochondria enzyme activity in experiments with rat nerve cells and human peripheral blood mononuclear cells [252,253]. 

Experiments with rats fed with AlCl_3_ via drinking water showed that Al^3+^ caused a reduction in the cytoplasm ATP level, a decrease in CI–CIV activity, and disturbance to mitochondrial DNA transcripts, which was followed by the inhibition of the mRNA expression of NADH dehydrogenases 1 and 2, cytochrome b, cytochrome c oxidase subunits 1 and 3, and ATP synthase [254]. By violating the mitochondrial energy metabolism, AlCl_3_ caused an increase in important liver aspartate and alanine aminotransferases and histopathological lesions. Aluminum nanoparticles (AlNPs), more so than Al^3+^, induced apoptosis, a cellular viability decrease, a DNA damage increase, mitochondrial dysfunction, and oxidative stress, as well as a decrease in ΔΨ_mito_ and reduced glutathione, in experiments with human hepatic HepG2 cells and differentiated HepaRG cells [255].

Ga^3+^ ions transport into cells via the use of cell Fe^3+^ transport systems [256]. A decrease in Ga^3+^-induced cellular iron uptake and intracellular iron homeostasis disruption resulted in mitochondrial function inhibition due to an intracellular ROS production increase and a reduced glutathione decrease as well as iron-containing proteins being targeted in the electronic transport chain [256]. Ga^3+^ triggers apoptosis by *Bax* or *p53* activation, which was followed by translocation to mitochondria, which results in ΔΨ_mito_ loss, an ROS production increase, and cytochrome c release into the intermembrane space and the cytoplasm [256]. 

Indium(III) compounds were found to induce a cell viability decrease, a lipid peroxidation and ROS production increase, reduced glutathione depletion, an SOD activity decrease, ΔΨ_mito_ decline, and oxidative stress damage as well as the expression of apoptotic genes (p53, bax, bcl-2, caspase-3, and caspase-9), oxidative DNA damage, and the formation of condensed chromosomal bodies in experiments in vitro with human lung epithelial A549 cells, lung-derived epithelial (LA-4) cells, human bronchial epithelial BEAS-2B cells, human embryonic kidney 293T cells, and mouse monocyte macrophage RAW 264.7 cells [243,257,258,259,260,261]. In^3+^ in experiments with human GM5565 skin fibroblasts affected mitochondrial morphology, and organelles became fragmented with a dotted geometry, which may indicate organelle aggregation caused by increased ROS production [262]. Indium oxide nanoparticles were found to have an acute toxic influence on epithelial 16HBE and macrophage (RAW264.7) cells in experiments with rats subjected to particle inhalation [263]. Damage to mitochondria, a rough endoplasmic reticulum, and a cell viability decrease were found. 

These trivalent cations (Al^3+^, Ga^3+^, and In^3+^) enter cells using Fe^3+^ transport systems, which was followed by ROS production with lipid and protein oxidation and iron homeostasis disruption and a reduced glutathione decrease. As a result, damage to iron-containing proteins takes place in the electronic transport chain. There is an entirely different mechanism for inducing both oxidative stress (Figure 1) and apoptosis (Figure 2). The free radical oxidation of thiol groups caused by these metals leads to decreased CI–CIV activity, ATP production, and ΔΨ_mito_ decline as well as decreased SOD and catalase activities. The expression of apoptotic genes and oxidative DNA damage accompanies apoptosis induction. Of course, these trivalent metals can be attributed to toxicants disrupting iron and calcium metabolism. This circumstance can harm the hematopoietic system and the nervous system cells. However, accumulating these metals in the body can aggravate diseases associated with the degradation of the nervous and hematopoietic systems and the synthesis of hemoglobin and other iron-containing proteins. 

**As(III). *Mitochondrial research.*** As(III) (As_2_O_3_ or AsO_2_^−^) induced a decrease in CI and CII activities, ΔΨ_mito_, and glutathione content, as well as an increase in ROS production, lipid peroxidation, cytochrome c release, and swelling, in isolated RLM or mouse liver mitochondria [264,265,266,267]. Some of these As^3+^ toxic effects were increased by pyruvate [264] and attenuated by alpha lipoic and ellagic acids [265,268]. AsO_2_^−^ induced an H_2_O_2_ production increase in isolated RHM [269]. 

As_2_O_3_-induced MPTP opening was accompanied by an increase in mitochondrial swelling and Ca^2+^-induced Ca^2+^ release, cytochrome c release from the matrix in the inner membrane space, and ΔΨ_mito_ decline in succinate-energized RLM [270,271]. However, Ca^2+^ is required for MPTP opening. CsA, NEM, or RR inhibited MPTP. As^3+^ did not affect the mitochondrial GSH content in RLM [271]. RLM takes up H_2_AsO_4_^−^ via the Pi-dependent pathway and can reduce As^5+^ to As^3+^ in participating thioredoxin reductase and reduced glutathione, which is followed by the extrusion of As^3+^ from mitochondria [272]. This process is more effective with CI substrates than succinate or ADP but is abolished by electron transport inhibitors, uncouplers, ATP synthase inhibitors, and Pi-transport inhibitors. 

**As(III). *Cell research.*** The available literature analysis makes it possible to attribute As(III) chemical forms (As_2_O_3_, AsO_2_^−^) to mild thiol reagents. This is the reason for their widespread use in ancient Chinese medicine and in the treatment of certain cancers. Arsenic (As_2_S_2_, As_2_O_3_, NaAsO_2_, and Na_2_HAsO_4_) and mercury (HgS, HgCl_2_, and MeHg) are widely used in oriental medicine [273]. It was found that the AsO_2_^−^-induced superoxide increase in human myeloid leukemia U937 cells has two contrary pathways: the first leads to the promotion of Nrf2 signaling followed by GSH biosynthesis; the second results in DNA damage and MPTP opening with rapid and massive apoptotic cell death [274,275]. As(III) (AsO_2_^−^ and As_2_O_3_) increased the intracellular Ca^2+^ concentration and mitochondrial dysfunction due to Ca^2+^ mobilizing from the ER in human myeloid leukemia U937 cells, human bronchial epithelial HBE cells, and human umbilical and bone marrow mesenchymal stem cells [276,277,278].

As(III) (As_2_O_3_ and AsO_2_^−^) induced apoptosis with the expression of apoptosis-inducing factors (*Bcl-2* and *Bcl2*), which was followed by a cell viability decrease, the activation of some caspases, and cytochrome c release in the cytoplasm in experiments with L02 hepatocytes, Hepa-1c1c7 cells, Hela cells, mouse insulinoma MIN6 cells, isolated rat hepatocytes, adult rat hepatic stem cells, H9c2 cardiomyocytes, human hepatocellular carcinoma HepG2 cells, colon cancer HT-29 cells, human myeloid leukemia U937 cells, human bronchial epithelial HBE cells, and human umbilical and bone marrow mesenchymal stem cells [161,162,271,276,277,278,279,280,281,282,283,284,285,286,287,288]. However, As(III) (As_2_O_3_ and AsO_2_^−^) brought about oxidative stress. The stress manifested in MPTP opening, mitochondrial swelling, ΔΨ_mito_ decline, a decrease in ATP, reduced glutathione, an ROS production increase, and cytochrome c release in investigations with PC12 cells, Hela cells, isolated rat hepatocytes, rat cardiomyocyte H9c2 cells, L02 hepatocytes, lung bronchial epithelial HBE cells, Hela cells, mouse insulinoma MIN6 cells, laryngeal squamous carcinoma Hep-2 cells, Chang human hepatocytes, mouse embryonic fibroblasts, mouse hepatocytes, human hepatocellular carcinoma HepG2 cells, colon cancer HT-29 cells, cervical cancer cells, primary cultured mouse hepatocytes, and human myeloid leukemia U937 cells [161,162,271,279,281,282,283,284,285,287,288,289,290,291,292,293,294,295,296]. In addition, AsO_2_^−^ in L02 hepatocytes induced ERK signaling to activate both apoptosis and mitophagy [282] along with ferroptosis in mouse insulinoma MIN6 cells or PC-12 cells [283,297]. Oxygen consumption rates and CI and catalase activity without/with FCCP were increased with low AsO_2_^−^ but decreased with high AsO_2_^−^ in experiments with human and mouse primary hepatocytes [293]. Diamide or PAO but not *t*BHP or AsO_2_^−^ caused MPTP opening due to the oxidation of ANT critical Cys^56^ [298]. 

N-acetyl-L-cysteine, L-ascorbic acid, and selenite (SeO_3_^2−^) prevented AsO_2_^−^-induced apoptosis, oxidative stress, a cell viability decrease, mitochondrial disfunction, cytochrome c release, and an ROS production increase in experiments with L02 hepatocytes, mouse oligodendrocyte precursor cells, human myeloid leukemia U937 cells, and acute promyelocytic leukemia NB4 cells [279,299,300,301]. The effect of SeO_3_^2−^ can be caused by the inhibition of AsO_2_^−^ transport in NB4 cells [301]. Pterostilbene or tert-butylhydroquinone activating the Nrf2 pathway alleviated similar AsO_2_^−^-induced effects in human HaCaT keratinocytes, mouse epidermal JB6 cells, and human epithelial HaCaT cells [302,303]. An increased glutathione biosynthesis capacity prevented As^3+^-induced apoptosis due to a decrease in caspase activation and cytochrome c release in the cytoplasm in mouse liver hepatoma Hepa-1c1c7 cells [280]. Autophagy can inhibit As_2_O_3_-induced apoptosis in HL60 cells in the initiation stage [304]. Metallothionein or methyl donors (betaine, methionine, and folic acid) alleviated the As^3+^-induced ΔΨ_mito_ decline and ROS production increase in PC12 cells or isolated rat hepatocytes [289,290]. The ferroptosis inhibitor Fer-1 attenuated the similar toxic effects of As_2_O_3_ on rat cardiomyocyte H9c2 cells [291]. Experiments with rats co-exposed to NaAsO_2_ and NaF showed oxidative stress and autophagy in myocardial tissue and cells [287]. As^3+^-induced oxidative stress and the apoptosis mitochondrial pathway were alleviated in experiments with the glutathione treatment of chronic arsenic-exposed mice [305]. 

Arsenic nanoparticles induced by apoptosis, mitochondrial swelling, ΔΨ_mito_ decline, a glutathione decrease, an ROS production increase, and membrane integrity were disturbed in isolated rat hepatocytes [306]. 

**Sb(III). *Mitochondrial and cell research.*** Experiments with Sb^3+^-treated mitochondria isolated from A549 cells showed an increase in SOD activity and a decrease in CI and CIII activities, glutathione peroxidase, glutathione reductase, and thioredoxin reductase [307]. The compounds and complexes of pentavalent and trivalent antimony with potassium antimonyl tartrate are used to treat leishmaniasis caused by protozoan parasites [308,309,310]. Sb(III) (Sb_2_O_3_ and SbCl_3_) induced apoptosis and cell viability loss; an increase in ROS production; and a decrease in the cytoplasm glutathione level, ΔΨ_mito_, and ATP content in human lung adenocarcinoma A549 cells, human embryonic kidney HEK-293 cells, and CCRF-CEM line cells [307,311,312]. Potassium antimonyl tartrate (K_2_[SbC_2_H_2_(O)_2_(COO)_2_]_2_) and As_2_O_3_ had similar effects in experiments with human lymphoid tumoral cells and human myeloid leukemic HL60 cells [313,314]. Sb^3+^-induced apoptosis in CCRF-CEM cells increased in the presence of both buthionine sulfoximine (a gamma-glutamylcysteine synthetase inhibitor) and sodium ascorbate, which reduce intracellular glutathione levels in human myeloid leukemic HL60 cells [312,314]. The metal-induced ROS production increase and cell viability decrease showed the order of As^3+^ > As^5+^ > Sb^3+^ > Sb^5+^, and these effects were the main cause of Sb cytotoxicity in these cells [311]. 

As(III), similar to the other heavy metals above, induced oxidative stress (Figure 1) accompanied by a decrease in CI and CII activities, MPTP opening, an ROS production increase, and an ATP production and ΔΨ_mito_ decline. However, As^3+^ (unlike the heavy metals discussed above) can be classified as a mild thiol effector due to its weak influence on mitochondrial GSH content. As(III) similar to Pb^2+^ increases cytoplasm [Ca^2+^] mobilizing Ca^2+^ from the ER, which is followed by mitochondrial dysfunction due to the calcium overload of those organelles. That elimination of the As(III)-induced oxidative stress and apoptosis consequences in the presence of NAC, ascorbate, and selenite is probably due to the significant reversibility of these As(III)-induced processes. It should be noted that this circumstance is indicated by preventing As^3+^-induced apoptosis due to the acceleration of the processes of glutathione biosynthesis or the action of metallothioneins or methyl donors on cells. A few studies on Sb(III) compounds found effects similar to those of As(III). Undoubtedly, the moderate binding of As(III) to the thiol groups of various proteins and enzyme complexes has become the reason for the recent rapid growth in research on the search for and synthesis of arsenic-containing drugs that selectively suppress the growth of cancer cells for the treatment of oncological diseases.

**Cr(VI). *Mitochondrial research.*** Cr_2_O_7_^2−^ induced a state 3 and 3U_FCCP_ respiration decrease, an ROS production increase, and ΔΨ_mito_ decline in isolated RLM energized by CI and CII substrates [315,316]. However, state 4 respiration increased and peaked at 250 μM K_2_Cr_2_O_7_, which was followed by a sharp decrease in respiration [316]. Dichromate weakly affected the inner membrane passive proton permeability; however, the FCCP-induced permeability of the IMM was partially reduced by Cr_2_O_7_^2−^ [316]. RLM took up Cr(VI), which reduced further to Cr(V) in the matrix [317]. Glutathione eliminated these effects of Cr_2_O_7_^2−^ [81,315]. Cr_2_O_7_^2−^ induced the state 3 inhibition and ΔΨ_mito_ decline in mitochondria isolated from L02 hepatocytes [318].

**Cr(VI). *Cell research.*** Cr_2_O_7_^2−^ caused caspase-3 dependent apoptosis, and mitochondrial dysfunction was accompanied by a cell viability decrease, MPTP opening, mitophagy, a ΔΨ_mito_ decline, energy metabolism disturbance with an ATP production decrease, cytochrome c release, a CI–CIII activity decrease, and an intracellular Ca^2+^ increase due to a mitochondrial ROS production increase and endoplasmic reticulum (ER) stress, followed by Ca^2+^ release, in experiments with L02 hepatocytes [318,319,320,321,322,323,324,325,326,327]. Cr(VI) (CrO_4_^2−^ and Cr_2_O_7_^2−^) induced mitochondrial apoptosis, and this was dependent on both caspase-3 with a preceding p53 signal and mitophagy. Oxidative stress, Ca^2+^ overload (due to mitochondrial and ER stress), MPTP opening, a cell viability decrease, CI and CII inhibition, a mitochondrial ATP production decrease, ΔΨ_mito_ decline, a mitochondrial swelling increase, cytochrome c release, and an ROS production and lipid peroxidation increase were also observed here in experiments with mouse SSCs/progenitors, human lung A549 carcinoma epithelial cells, human lung H1299 adenocarcinoma cells, human neuroblastoma SH-SY5Y cells, human bronchial epithelial BEAS-2B cells, cultured cerebellar granule neurons, human hepatoma Hep3B cells, rat granulosa cells, mouse epidermal Cl 41 cells, human bronchoalveolar carcinoma H358 cells, and human myeloid leukemia U937 cells [321,328,329,330,331,332,333,334,335,336,337,338,339]. A Cr_2_O_7_^2−^ induced decrease in Ca^2+^, Mg^2+^-ATPase and Na^+^, K^+^-ATPase activities was found in human hepatoma Hep3B cells [321]. Some of these Cr_2_O_7_^2−^-induced effects were inhibited by N-acetyl-L cysteine, vitamin C, BAPTA-AM, and clusterin (disulfide-linked multifunctional glycoprotein). Zn^2+^ attenuated the CrO_4_^2−^-induced mitochondrial caspase-3 apoptosis and oxidative stress that manifested as ΔΨ_mito_ decline and an ROS or H_2_O_2_ production increase in human tumor Hep-2cells [340].

Cr(VI) compounds are of industrial origin and very rarely occur in nature in the form of crocoite mineralade, PbCrO_4_. Being the most potent oxidizing agent, Cr_2_O_7_^2−^ induced oxidative stress (Figure 1) in addition to lysosomal membrane rupture, GSH oxidation, CI and CII inhibition, a decline in ΔΨ_mito_ and ATP production, MPTP opening, ROS production, and an increase in lipid peroxidation in experiments with various cells in vitro. At the same time, under acting Cr(VI) compounds, as in the case of the heavy metals considered above, apoptosis (Figure 2) was induced due to caspase-3 activation with a preceding p53 signal and mitophagy. Antioxidants and ROS scavengers prevented these cytotoxic effects. Cr(VI) toxicity may increase due to the activation of nitrosylation processes in cells, which was followed by the activation of apoptosis by a mechanism involving S-nitrosylation and nitric oxide-dependent stabilization of the Bcl-2 protein [341]. Therefore, cell damage caused by Cr(VI) compounds can provoke carcinogenesis, leading to oncological diseases, including lung cancer, due to the malignant transformation of human lung epithelial cells.

**U(VI). *Mitochondrial and cell research.*** UO_2_^2+^ attenuated mitochondrial respiration in states 3 and 3U_DNP_ and induced both a H_2_O_2_ formation increase and a slight calcium retention increase in succinate-energized RLM [342]. UO_2_^2+^ induced an ROS production and lipid peroxidation increase as well as a reduced glutathione content and ΔΨ_mito_ decrease in the isolated kidney mitochondria of rats injected with uranyl acetate [343]. The toxicity of UO_2_^2+^ manifested as CII and CIII inhibition, mitochondrial swelling, ΔΨ_mito_ decline, and an increase in H_2_O_2_ and ROS production and lipid peroxidation as well as a decrease in cell viability, reduced glutathione content, mitochondrial ATP content, the ATP/ADP ratio, cytochrome c release, and outer mitochondrial membrane damage in experiments with primary rat hepatocytes, human dermal fibroblast primary cells, and isolated rat kidney mitochondria (RKM) energized by CI and CII substrates [343,344,345,346]. MPTP inhibitors (carnitine, CsA, and trifluoperazine), antioxidants (catalase), ROS scavengers (mannitol and DMSO), or a U^6+^ reduction inhibitor (Ca^2+^) prevented the UO_2_^2+^-induced ΔΨ_mito_ decrease and lysosomal membrane hepatocyte damage [344]. Ca^2+^ blocked the glutathione or cysteine capacity to reduce U^5+^ and U^6+^ to U^4+^ to form O_2_^•−^ in these hepatocytes [344]. Beta-glucan and butylated hydroxyl toluene prevented both these toxic UO_2_^2+^ effects and the UO_2_^2+^-induced outer mitochondrial membrane damage in RKM [347]. 

Uranyl binding with cyt b_5_, cyt c, and the cyt b_5_–cyt c complex may indicate that UO_2_^2+^-induced apoptosis forms a dynamic cyt b_5_–cyt c complex [348]. UO_2_^2+^ induced apoptosis with the activation of caspases-3, -8, -9, and -10 to indicate a mitochondria-dependent signaling pathway in rat kidney NRK-52E proximal cells, rat hepatic BRL cells, HEp-2 cells, and human dermal fibroblast primary cells [342,346,349,350]. There was also UO_2_^2+^-induced oxidative stress accompanied by a cell viability decrease, an ROS production increase, a lipid peroxidation increase, a reduced glutathione content decrease, and ΔΨ_mito_ collapse. Zn^2+^ (an effective heavy metal poisoning antidote) protected against UO_2_^2+^-induced apoptosis in human kidney HK-2 cells [351]. In addition, Zn^2+^ prevented the UO_2_^2+^-induced cell viability decrease, cytochrome c release to the cytoplasm, ΔΨ_mito_ decline, lactate dehydrogenase (LDH) depletion, and increase in ROS production and catalase and glutathione concentrations [351]. 

UO_2_^2+^ (similar to heavy metals) initiated apoptotic processes with the activation of caspases-3 and -9 and formed a dynamic cyt b5–cyt c complex. Moreover, uranyl-induced mitochondrial oxidative stress (Figure 1) manifested as an ROS production and lipid peroxidation increase, a GSH decrease, CII and CIII inhibition, MPTP opening, and ΔΨ_mito_ decline. The cytotoxic effects of UO_2_^2+^ were markedly attenuated in experiments with MPTP inhibitors, antioxidants, and ROS scavengers. High uranyl toxicity was found for some bacteria [352]. In addition, uranyl disrupts cellular calcium metabolism. One can hypothesize that uranyl toxicity (together with natural radioactivity) may manifest itself through disturbances in calcium-dependent metabolic processes, exacerbating metabolic disorders. The toxicity of UO_2_ was higher than the toxicity of Al and comparable to the toxicity of Cu, Zn, and Pb, but it was lower than the toxicity of Cd and Ag [353]. 

**Modified heavy metals. *Cell research.*** Reduced glutathione was shown to react directly with Ag^+^ ions [94]. Tl^3+^ ions unlike Tl^+^ ones oxidized glutathione [354]. The toxic effects of MeHg were more potent than those of Hg^2+^ on zebrafish ZF4 and human gingival fibroblasts [153,163]. The co-exposure of AgNPs, Cd^2+^, and Hg^2+^ showed increased toxicity in experiments with human hepatocarcinoma HepG2 cells, with AgNP + Hg^2+^ being less toxic than AgNPs + Cd^2+^ [355]. Nanoparticles (CdO or Ag) inhibited succinate dehydrogenase in BRL-3A hepatocytes [104]. Nano-carbon black with Pb^2+^ particles simulating the atmosphere fine particles formed a special complex that activated apoptotic signaling pathways, impaired ΔΨ_mito_, and inhibited the lysosomal function in rat alveolar macrophages [356]. Pb^4+^ as Pb(acetate)_4_ increased intracellular human neuroglobin and ROS production in breast cancer MCF-7 cells and induced mitochondrial apoptosis [357]. Trialkyllead compounds similar to Ca^2+^ with valinomycin induced an increase in mitochondrial swelling, state 4 respiration, and K^+^ efflux, as well as a decrease in state 3 respiration, in succinate-energized RLM injected in KCl but not in a KNO_3_ medium [358,359,360]. Minerals containing heavy metals with a maximum oxidation state (Tl^3+^ and Pb^4+^) do not occur in nature, and their use in research is purely for laboratory use. Ph_3_Sb(V)O in complex with carvacrol was found to reduce in participating glutathione to a toxic form containing a Ph_3_Sb(III) complex that resulted in apoptosis, mitochondrion damage, and mitochondrial membrane permeabilization in human breast adenocarcinoma MCF-7 cells [361]. The AlF_4_^−^ complex being isomorphous to Pi inhibited beef heart mitochondrial F_1_-ATPase, pig kidney Na^+^, K^+^-ATPase, plasmalemmal stomach smooth muscle Ca^2+^, and Mg^2+^-ATPase [362,363,364]. Gold(I)-derived complexes synthesized on the basis of Au[P(Ph)_3_]^+^Cl^−^ were proven to be effective trypanothione reductase inhibitors and proposed to treat leishmaniasis [365]. These gold complexes induced oxidative stress with an ROS production increase, mitochondrial damage, and a mitochondrial respiration decrease in experiments with human monocyte-derived macrophage THP-1 cells infected by leishmaniasis parasites. 

## 3. Conclusions

This review analyzes the causes and consequences of apoptosis and oxidative stress that occur in mitochondria and cells exposed to the toxic effects of different valent heavy metals (Ag^+^, Tl^+^, Hg^2+^, Cd^2+^, Pb^2+^, Al^3+^, Ga^3+^, In^3+^, As^3+^, Sb^3+^, Cr^6+^, and U^6+^). Experiments with different cells and mitochondria showed that the heavy metals under review induced apoptosis characterized by caspase-3 and -9 activation, *Bax* and *Bcl-2* expression, and mitogen-activated protein kinases (ERK, JNK, p53, and p38). Reduced cell viability and oxidative stress were observed, and they manifested as MPTP opening, mitochondrial swelling, an ROS and H_2_O_2_ production increase, lipid peroxidation, cytochrome c release, and a reduced glutathione and oxygen consumption decrease as well as cytoplasm and matrix calcium overload due to Ca^2+^ release from ER. There was also an ATP synthesis decrease and ΔΨ_mito_ decline due to CI–CIII dysfunction found in experiments in vitro with both cells and mitochondria. This dysfunction is due to the interaction of some metals (Ag^+^, Hg^2+^, Cd^2+^, Pb^2+^, As^3+^, and Sb^3+^) with high-affinity thiol groups of the respiratory complexes and adenine nucleotide translocase. On the one hand, other metals (Al^3+^, Ga^3+^, In^3+^, Cr^6+^, and U^6+^) induce H_2_O_2_ and ROS production as well as lipid peroxidation that results in the function oxidative damage of these respiratory complexes. Some metals (Pb^2+^, Al^3+^, and Ga^3+^) disturb Fe^2+^ metabolism and distort the structure of the iron–sulfur centers of the mitochondrial respiratory chain. At the same time, it should be noted that Tl^+^ among the metals under review possesses significant differences. On the one hand, the toxic effects of Tl^+^ and other heavy metals showed an obvious similarity in experiments with cells. On the other hand, Tl^+^ in experiments with isolated mitochondria did not induce MPTP opening and showed negligible reactions with mitochondrial thiol groups and no inhibition of respiratory enzymes. The toxic effects of Tl^+^ were similar to those of heavy metals only in experiments with calcium-loaded mitochondria. Thus, the similarity of the effects of Tl^+^ and other heavy metals may be due to the increased cytoplasmic calcium concentration induced by these metals. However, the toxicity of thallium being greater than the toxicity of mercury, lead, cadmium, copper, and zinc to humans may be due to the fact that the existing metallothionein-dependent mechanisms do not reduce thallium toxicity, which is in contrast to the toxicity of these heavy metals. Another reason for the thallium toxicity may be our hypothesized decrease in reduced glutathione in the matrix as a result of the reversible oxidation of Tl^+^ to Tl^3+^ near the centers of the generation of ROS in the respiratory chain. 

## Figures and Tables

**Figure 1 ijms-24-14459-f001:**
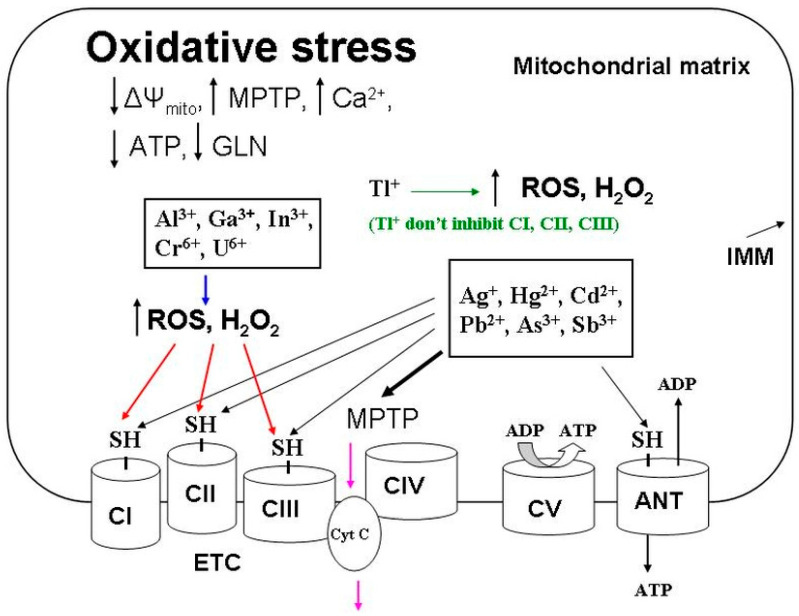
General elements for toxic mechanisms of inducing mitochondrial oxidative stress via heavy metals. Heavy metals (Ag^+^, Hg^2+^, Cd^2+^, Pb^2+^, As^3+^, and Sb^3+^) induce oxidative stress by blocking respiratory complexes with MPTP opening due to interacting with the thiol groups shown in the picture. Other metals (Tl^+^, Al^3+^, Ga^3+^, In^3+^, Cr^6+^, and U^6+^) inactivate these thiol groups by indirectly oxidizing them due to activation of the production of oxygen radicals (ROS and H_2_O_2_). Black arrows show the metal’s reaction with thiol groups (SH-). The MPTP induction by heavy metals is shown by the bold arrow. The induction of ROS and H_2_O_2_ is marked with a blue arrow. The ROS-induced oxidation of thiol groups is shown by red arrows. Pink arrows show the cytochrome c release into the intermembrane space due to the MPTP opening. Abbreviations: ADP, adenosine diphosphate; ANT, adenine nucleotide translocase; ATP, adenosine triphosphate; CI–CV, inner mitochondrial membrane complexes I–V; Cyt C, cytochrome C; ETC, electron transport chain; GSH, reduced glutathione; IMM, inner mitochondrial membrane; MPTP, mitochondrial permeability transition pore; ROS, reactive oxygen species; SH-, molecular thiol groups; ΔΨ_mito_, inner membrane potential.

**Figure 2 ijms-24-14459-f002:**
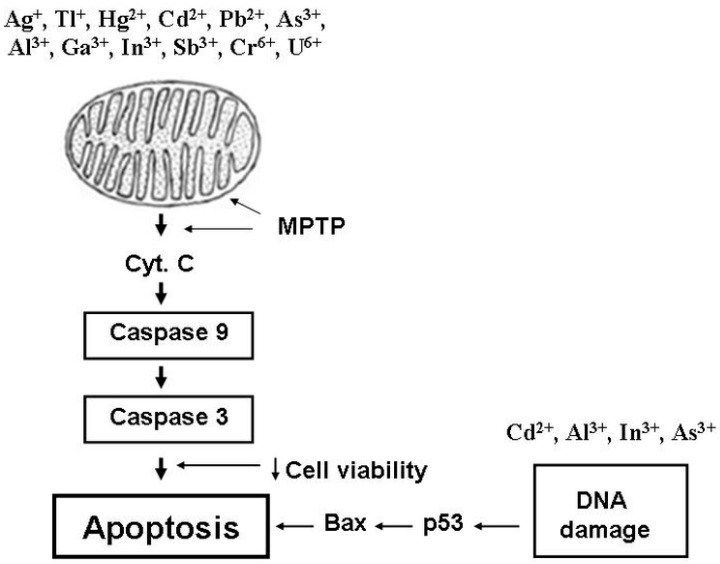
General links in mitochondria-dependent pathways of intracellular apoptosis induction by different-valence heavy metals. Heavy metals (Ag^+^, Tl^+^, Hg^2+^, Cd^2+^, Pb^2+^, As^3+^, Sb^3+^, Al^3+^, Ga^3+^, In^3+^, Cr^6+^, and U^6+^) induce mitochondrial permeability transition pore (MPTP) opening, which is accompanied by cytochrome c release into the intermembrane space. As a result of this process, caspases-3 and -9 are activated and trigger apoptosis induction. Some metals (Cd^2+^, As^3+^, Al^3+^, and In^3+^) damage DNA, resulting in bax and p53 activation that induces apoptotic processes. Arrows show MPTP induction. Bold vertical arrows show the sequence of events associated with mitochondria during the induction of intracellular apoptosis by heavy metals. Apoptosis induction pathways associated with DNA damage are indicated by horizontal arrows. Abbreviations: Cyt. C, cytochrome C; MPTP, mitochondrial permeability transition pore.

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
