# Peer review of "Mitochondrial Oxidative Stress Is the General Reason for Apoptosis Induced by Different-Valence Heavy Metals in Cells and Mitochondria"

_ijms, 2023, doi:10.3390/ijms241914459_

Round 1
Reviewer 1 Report
Point by point
Manuscript: The mitochondrial oxidative stress is general reason of apoptosis induced by different valence heavy metals on cells and mitochondria
Article Type: Review
In the present article the author summarizes the existing information about mitochondrial damage generated after direct addiction (to mitochondria) or to cell lines of different heavy metals Ag+, Tl+, Hg2+, Cd2+, Pb2+, Al3+, Ga3+, In3+, As3+, Sb3+, Cr6+, U6+). The author effectively summarizes the bioenergetics and redox damages caused by these metals in mitochondria, focus in the deregulation of GSH and the damage of thiols present in the mitochondrial proteins involved in energy metabolism. This triggers the OXPHOS efficiency loss, the increase in oxidant stress, calcium overload and the opening of MPTP favoring the induction of cell death mediated by caspases. Although the review is interesting, well prepared and provides a good description of the alterations at the bioenergetics level, in several sections it lacks of integration that allows us to glimpse the molecular mechanisms associated with the damage, as well as the relevance at the pathophysiological level of mitochondrial alterations. described. Therefore, somel corrections are recommended before being published.
Major points:
• Although the description of the bioenergetic and the respiratory states alterations triggered by each metal is enough. Their integration into molecular mechanisms for each metal is not present in most of the sections. Therefore, a concluding paragraph for each metal addressed or an integrative figure is necessary.
• Heavy metal poisoning is a process widely described in the literature, in whose pathophysiology the induction of cell sample plays a fundamental role. Therefore, given the key role of mitochondria as the central axis in the intrinsic apoptotic pathway, the role of mitochondria in the pathophysiology of these diseases should be at least mentioned, in addition to being contextualized with the information given in the article.
• In the same way, bioenergetic alterations such as the decrease in respiratory states and the induction of ROS production favor the development of different diseases, this should be discussed when describing the mitochondrial dysfunction generated by the corresponding metals, such as calcium overload or transition pore opening role in the pathologies.
• The figure legends of the diagrams are scarce and lacks the explanation of the illustrated mechanisms, as well as the definitions of the necessary abbreviations.
• Because the importance of alterations in mitochondrial GSH and in the redox state of the thiols of the proteins of the mitochondrial complexes, it is suggested to discuss the possible role post translational modification such as s-nitrosylation or s-glutathionylation in the electron transport system mitochondrial damage
Minor points:
• A large number of acronyms are not defined, or the definition appears later in the text. A deep review of acronyms and their definitions is necessary
• The English and the grammar of the text should be checked, since there are several typos
• The English and the grammar of the text should be checked, since there are several typos
Author Response
Special Issue Editor IJMS
Prof. Dr. Toshiyuki Kaji
Dear Professor Kaji,
There I present my replies to the two reviewer' comments on my manuscript "The mitochondrial oxidative stress is general reason of apoptosis induced by different valence heavy metals on cells and mitochondria". Despite a certain share of criticism, I am very grateful to these reviewers because of these remarks were very useful to improve my manuscript. Across new version our manuscript my corrections are highlighted in green (for 1st reviewer) and yellow (for 2nd revievwer). Please find our point by point response to each of the comments.
Sergey Korotkov, PhD in Biochemistry and MS in Chemistry
September 8, 2023
St. Petersburg
My Responses to Reviewer 1.
Major points:
- Although the description of the bioenergetic and the respiratory states alterations triggered by each metal is enough. Their integration into molecular mechanisms for each metal is not present in most of the sections. Therefore, a concluding paragraph for each metal addressed or an integrative figure is necessary.
A discussion of the problem of integration of the heavy metals considered in this review is added at the end of each section with the exception of the section on chemical derivatives of some heavy metals.
- Heavy metal poisoning is a process widely described in the literature, in whose pathophysiology the induction of cell sample plays a fundamental role. Therefore, given the key role of mitochondria as the central axis in the intrinsic apoptotic pathway, the role of mitochondria in the pathophysiology of these diseases should be at least mentioned, in addition to being contextualized with the information given in the article.
At the end of each section there is a mention of how toxic damage to mitochondria can lead to diseases caused by heavy metal poisoning or aggravate existing chronic diseases.
- In the same way, bioenergetic alterations such as the decrease in respiratory states and the induction of ROS production favor the development of different diseases, this should be discussed when describing the mitochondrial dysfunction generated by the corresponding metals, such as calcium overload or transition pore opening role in the pathologies.
A discussion of the role of mitochondrial calcium overload and concomitant MPTP opening has been added at the end of the sections.
- The figure legends of the diagrams are scarce and lacks the explanation of the illustrated mechanisms, as well as the definitions of the necessary abbreviations.
The necessary explanations were added in the figure legends of the common mechanisms to these metals. The definitions of the necessary abbreviations were added in the text.
- Because the importance of alterations in mitochondrial GSH and in the redox state of the thiols of the proteins of the mitochondrial complexes, it is suggested to discuss the possible role post translational modification such as s-nitrosylation or s-glutathionylation in the electron transport system mitochondrial damage
The possible role of s-nitrosylation or s-glutathionylation in the toxic effect of some metals showed on pages 19 and 29.
Minor points:
- A large number of acronyms are not defined, or the definition appears later in the text. A deep review of acronyms and their definitions is necessary
Abbreviations and their definitions are presented throughout the text of the review. A list of abbreviations has also been added at the end of the text.
- The English and the grammar of the text should be checked, since there are several typos
I checked English and the grammar of the text.

Reviewer 2 Report
The manuscript “The Mitochondrial Oxidative Stress Is General Reason of Apoptosis Induced by Different Valence Heavy Metals on Cells and Mitochondria” by Korotkov reviews the causes and consequences of apoptosis due to oxidative stress in mitochondria and cells after exposure to different valent heavy metals such as Ag+, Tl+, Cd2+ or Pb2+ among others. The author makes an exhaustive description of the effects of the different metals on oxidative stress increase which promotes apoptosis and mitochondrial dysfunction and discusses the differences in the toxic effects of Tl+ from other heavy metals.
The review was well conducted, the results are clearly presented, and the conclusions are supported by bibliography, but I have some comments:
- Some abbreviations are not defined. All of them should be described the first time they are used.
- The way of referring to respiratory complexes is not homogeneous. The author should use complex I, II, … instead of 1st and 2nd complexes.
- When talking about mitochondrial ATP synthase, the author sometimes writes F1Fo (which is the correct form) and others, F1F0.
- Superscripts are sometimes misspelled (for example in page 7 or in scheme 2).
- In page 14, there are some words highlighted.
English should be revised as there are some spelling and grammar mistakes.
Author Response
Special Issue Editor IJMS
Prof. Dr. Toshiyuki Kaji
Dear Professor Kaji,
There I present my replies to the two reviewer' comments on my manuscript "The mitochondrial oxidative stress is general reason of apoptosis induced by different valence heavy metals on cells and mitochondria". Despite a certain share of criticism, I am very grateful to these reviewers because of these remarks were very useful to improve my manuscript. Across new version our manuscript my corrections are highlighted in green (for 1st reviewer) and yellow (for 2nd revievwer). Please find our point by point response to each of the comments.
Sergey Korotkov, PhD in Biochemistry and MS in Chemistry
September 8, 2023
St. Petersburg
My Responses to Reviewer 2.
- “Some abbreviations are not defined. All of them should be described the first time they are used”.
Abbreviations and their definitions are presented throughout the text of the review. A list of abbreviations has also been added at the end of the text.
- “The way of referring to respiratory complexes is not homogeneous. The author should use complex I, II, … instead of 1st and 2nd complexes”.
The designations for the mitochondrial respiratory complexes I – complex V are respectively presented in the text as abbreviated terms (CI-CV).
- “When talking about mitochondrial ATP synthase, the author sometimes writes F1Fo (which is the correct form) and others, F1F0”.
This correct text form (F1Fo) is included throughout the text.
- “Superscripts are sometimes misspelled (for example in page 7 or in scheme 2)”.
The superscript in the text and Figure 1 (formerly Figure 2) has been corrected.
- “In page 14, there are some words highlighted”.
This text highlight has been removed.
